# Δ-ATTNMASK: ATTENTION-GUIDED MASKED HIDDEN STATES FOR EFFICIENT DATA SELECTION AND AUGMENTATION

## ABSTRACT

Visual Instruction Finetuning (VIF) is pivotal for post-training Vision-Language Models (VLMs). Unlike unimodal instruction finetuning in plain-text large language models, which mainly requires instruction datasets to enable model instruction-following ability, VIF also requires multimodal data to enable joint visual and textual understanding; therefore, it typically requires more data. Consequently, VIF imposes stricter data selection challenges: the method must scale efficiently to handle larger data demands while ensuring the quality of both visual and textual content, as well as their alignment. Despite its critical impact on performance, data selection for VIF remains an understudied area. In this paper, we propose Δ-AttnMask. This data-efficient framework quantifies sample quality through attention-guided masking of the model's hidden states, jointly evaluating image-text pairs without requiring domain labels, auxiliary models, or extra training. By computing loss differences (Δ) between the original states and states masked using high-attention regions, Δ-AttnMask intrinsically assesses sample quality. Experiments across multiple VLMs and datasets show that Δ-AttnMask achieves state-of-the-art performance with just 20% of data, accelerating training by **5×** while surpassing full-dataset baselines by +10.1% in overall accuracy. Its model-agnostic and data-agnostic design ensures broad applicability across modalities and architectures.

## 1 INTRODUCTION

Vision language models (VLMs) have made remarkable strides since their inception (Frome et al., 2013), evolving into practical tools for diverse applications such as visual question answering and reasoning (Shen et al., 2025), embodied intelligence (Ma et al., 2025), and scientific discovery (Bai et al., 2025). Built upon large language models (LLMs), VLMs extend LLMs to visual and textual understanding, enabling a richer comprehension of multimodal data. However, this enhanced capability comes at a cost, particularly during post-training. Visual instruction fine-tuning (VIF) is essential not only for instruction-following but also for aligning visual encoder outputs with the LLM backbone, which is a critical step for effective visual understanding. This dual objective of VIF process demands larger, more diverse datasets. For example, fine-tuning the LLM Vicuna-13B (Chiang et al., 2023) uses 70K samples, whereas when it is used in LLaVA (Liu et al., 2023) as a LLM backbone, the VLM necessitates 158K samples for satisfactory performance.

The ever-growing scale of vision–language datasets underscores the critical need for data-efficient learning, where both the quality and cross-modal alignment of visual and textual data substantially influence model performance. Among various strategies, data selection has emerged as a promising approach to accelerate training while maintaining or even enhancing performance (Yang et al., 2025b; Zhou et al., 2024; Wu et al., 2025). While data selection in single-modality settings typically targets the informativeness or diversity of representations in either the visual or textual domain, the scenario in VLMs is more complex. Effective data curation techniques for VLMs must consider the triadic interplay between images, associated text (e.g., captions), and task-specific labels. For instance, captions may omit key visual details, labels may not align with either modality, and cross-modal semantics can drift over large-scale datasets. These challenges complicate the assessment of data quality, as evaluating multimodal consistency requires joint reasoning over heterogeneous features and

metadata. Furthermore, the computational burden of such multi-modal analysis scales significantly with dataset size, necessitating efficient yet reliable metrics for cross-modal alignment. Addressing these difficulties is essential for advancing data-efficient learning in VLMs, where the goal is not merely to reduce dataset size but to retain the most semantically coherent and task-relevant examples.

Most existing methods may fall short of comprehensively addressing the challenges of large-scale, data-efficient learning in multimodal settings. TIVE (Liu et al., 2025) exhibit substantial performance degradation when applied to very large datasets, while ICONS (Wu et al., 2025) relies on expensive gradient computations, severely limiting scalability. Domain-specific filtering methods introduce additional constraints: (Xu et al., 2025) depends on external models whose biases may propagate into the selected dataset, and (Safaei et al., 2025) requires predefined data subdomains, reducing adaptability to new or evolving domains. LLM-specific techniques (Hu et al., 2025a; Jiang et al., 2025; Zhou et al., 2024; Xia et al., 2024; Li et al., 2024) are effective for purely textual corpora. These methods overlook cross-modality quality alignment, rendering them unsuitable for VLMs.

To address these limitations, we propose $\Delta$-AttnMask, a lightweight and effective data selection method that evaluates multimodal data quality directly from the model's internal responses during VIF to accelerate VLM training. Specifically, our method employs attention-score-guided masking: we selectively mask high-attention hidden states and measure sample alignment and quality efficiently in a single step by computing the loss difference between masked and unmasked samples. This brings two benefits: (1) it maintains low computational overhead by performing quality estimation in a single forward step, and (2) it does not rely on auxiliary models, handcrafted features, or additional annotations. Additionally, beyond selection, we explore its application in data augmentation to further enhance data effectiveness. Augmenting a high-quality 20% subset outperforms training on twice the raw data.

Extensive experiments across various VLMs, tasks, and datasets demonstrate that our approach effectively achieves lossless VLM training acceleration. Moreover, our method exhibit superior cross-architecture generalization across Qwen2-VL 2B, Qwen2-VL 7B (Wang et al., 2024b), and Llama-3.2-11B-Vision (Meta, 2024) across the MiniGPT-4 dataset (Zhu et al., 2023), the LLaVA Instruction 158K dataset from (Liu et al., 2023), and Vision Flan 191K from (Xu et al., 2023). In summary, our work makes three key contributions: 1). We propose $\Delta$-AttnMask, the first method to jointly assess visual-textual sample quality using only the model's reaction to the sample, requiring no auxiliary models or external resources. 2). Beyond selection, $\Delta$-AttnMask enables effective data augmentation. Reusing high-quality samples proves superior to doubling the dataset size. 3). On production-scale datasets and models, we validated our method. $\Delta$-AttnMask achieves at most 5× faster training and +10.1% accuracy gain using only 20% of data, showing its high potential in broad applicability in VLM post-training.

## 2 RELATED WORKS

The success of instruction finetuning in LLMs has inspired their adaptation to multimodal settings (Liu et al., 2023), enabling some modality-agnostic methods developed for LLMs to be applicable to VLMs as well. For example, there is work estimates data quality by comparing training loss to a holdout set (Mindermann et al., 2022). Xia et al. extend this idea by prioritizing training samples with gradients that are closely aligned with the downstream validation set (Xia et al., 2024). These methods underutilize available training resources and impose strict requirements on access to the target data distribution.

To reduce reliance on holdout or validation sets, alternative approaches have emerged. Works from Loshchilov et al. (Loshchilov & Hutter, 2016), Jiang et al. (Jiang et al., 2019), the GREATS by Wang et al. (Wang et al., 2024a), IFD by Li et al. (Li et al., 2024), and Jiang et al. (Jiang et al., 2025) employ loss or perplexity thresholds, assuming high-loss samples are most beneficial for LLM performance. However, such hard thresholding cannot distinguish between valuable data and noisy samples (Yang et al., 2025a). More critically, these methods, designed primarily for LLMs, lack explicit mechanisms to assess multimodal data quality or alignment.

Regarding data selection for VLMs, many existing works often overlook the importance of cross-modal alignment. For instance, Data Whisperer (Wang et al., 2025) evaluates image quality via text-attention scores in an in-context learning framework. The work from (Yang et al., 2025a) selects

data for CLIP (Radford et al., 2021) models by measuring similarity between image and caption labels. This approach is ill-suited for advanced VLMs that process both visual and textual inputs. Similarly, Bi et al. (Bi et al., 2025) introduce LLM selection inspirations by maximizing subset diversity via Pearson correlation between embeddings. Works from Yu et al. and Wang et al. (Yu et al., 2024; Wang et al., 2024c) refine this idea by incorporating criteria such as informativeness, uniqueness, and representativeness for individual modalities. Safaei et al. (Safaei et al., 2025) further enhance diversity through clustering and integrate subdomain weights computed by IFD to balance data mixing. Despite these advances, none comprehensively address the alignment between visual and textual inputs, their labels, and overall data quality.

Efficiency remains another major limitation of current methods. Xu et al. (Xu et al., 2025) depend on external VLMs to score image-text coherence, while Wu and Chen (Wu & Chen, 2025) combine CLIP-based scores with loss for selection. Liu et al. (Liu et al., 2025) compute per-sample influence scores, and Wu et al. (Wu et al., 2025) adjust it to score the influence of data to tasks, retaining only samples influential across multiple tasks. However, gradient-dependent influence scoring is computationally expensive. Works from Chen et al. (Chen et al., 2024) and (Wei et al., 2023) introduce additional overhead by training a separate model to weight samples based on CLIP-encoded or other features. These inefficiencies contradict the core accelerating training objective of data selection.

# 3 METHODOLOGY

## 3.1 OVERVIEW

$\Delta$-AttnMask quantifies the quality of visual-textual samples by measuring the model's sensitivity to attention-guided perturbations of its hidden states. The core idea is that high-quality samples exhibit greater loss degradation when critical regions of the input are masked. This principle can be illustrated through a straightforward variant of the method, such as directly masking image patches or text tokens. For low-quality inputs (e.g., blurry images or ambiguous instructions), introducing such noise has minimal impact on the model's output, resulting in a small change in loss between the original and masked conditions. We expect about equal high loss for both case. In contrast, for high-quality, semantically coherent samples, perturbing informative components leads to significantly different model interpretations, resulting in a substantial increase in loss.

By measuring this loss delta, i.e., $\Delta_i = \mathcal{L}_i^{\text{masked}} - \mathcal{L}_i$, and prioritizing samples with higher $\Delta_i$, we effectively identify a subset of high-quality, informative data for training. This strategy is directly supported by (Li et al., 2024), which demonstrates that the performance gap of a language model between with and without instructional context indicates data utility and can be leveraged for effective data selection in LLMs. Similarly, it has been established that a patch exerting significant influence on the network output exhibits higher sensitivity to perturbations (Shu & Zhu, 2019).

Formally, given a VLM $M$ and a dataset $\mathcal{D} = \{(x_i^v, x_i^t)\}_{i=1}^N$, where $x_i^v$ and $x_i^t$ denote the visual and textual inputs respectively, $\Delta$-AttnMask operates in three stages:

1. **Baseline Inference:** Compute the original loss $\mathcal{L}_i$ for each sample $(x_i^v, x_i^t)$ under the unmodified model.

2. **Attention-Guided Masking:** For each sample, identify high-attention hidden states in $x_i^t$ using the model's self-attention weights, mask the corresponding states in the output of transformer block or visual encoder, and recompute the loss $\mathcal{L}_i^{\text{masked}}$.

3. **Quality Scoring:** Assign a quality score $\Delta_i = \mathcal{L}_i^{\text{masked}} - \mathcal{L}_i$ to each sample. A larger $\Delta_i$ indicates higher data quality, reflecting the the sample contains crucial and helpful information that help the model to response as expected.

Eventually, samples with high $\Delta_i$ are prioritized during training, enabling more efficient learning from informative, well-aligned data.

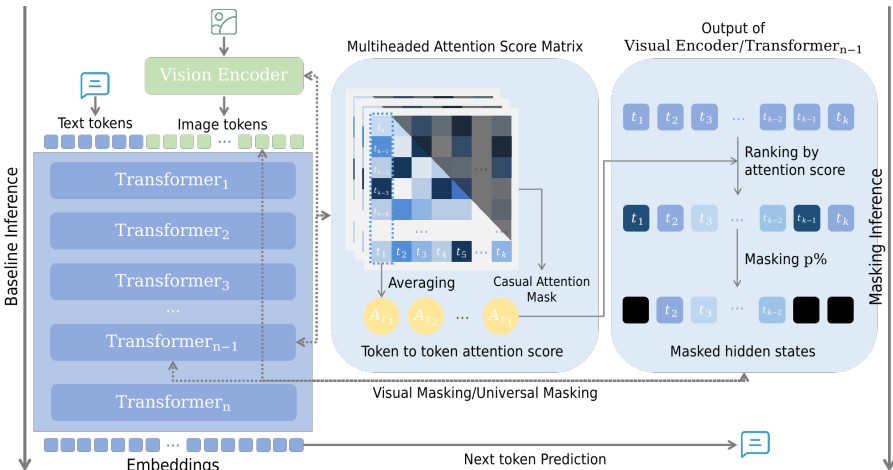

Figure 1: Overview of the Attention-Guided Masking mechanism, which follows a lightweight, two-step pipeline: (1) Compute the token-to-token average attention score across transformer layers; (2) Apply hidden state masking based on the attention score. In the figure, $n$ denotes the number of transformer blocks in the LLM backbone, $k$ represents the sequence length, and $p$ is the masking ratio.

## 3.2 MOTIVATION OF HIDDEN STATE MASKING

The direct masking of input approach introduced as an example in Section 3.1 faces practical limitations that hinder its direct application to data selection. For instance, randomly masking an image patch may fail to target semantically critical regions. For instance, in counting tasks, removing non-object areas yields negligible changes in model behavior. Moreover, this hard masking of raw input elements eliminates information completely and risks introducing artifacts unrelated to semantic content, making it difficult to capture fine-grained quality differences due to the model's overly challenging inference guessing.

To address these issues, we instead apply masking at the hidden state level, specifically within the transformer backbone. Hidden states aggregate contextualized representations across modalities and capture global semantics, making them more suitable for probing model sensitivity. By introducing the hidden state masking, we therefore avoid the risks and disadvantages of naive hard masking.

## 3.3 IMPLEMENTATION DETAILS

As for the specific masking target, we choose to avoid masking the output of the final transformer layer, as autoregressive generation naturally relies only on the last hidden state to predict the next token, masking here therefore has no impact on the prediction. Conversely, masking too early, such as before visual features are projected into the language model space, disrupts cross-modal integration, produces unstable signals, and causes the masking to collapse into variations similar to hard masking.

We introduce two variants of $\Delta$-AttnMask based on different masking strategies. While the dual-masking strategy achieves state-of-the-art (SOTA) performance, we further optimize and simplify it, achieving slightly lower yet SOTA performance, but tremendously reducing computation by 33%.

We initially explored a dual-masking approach: separately masking visual and textual hidden states in the visual encoder and the LLM backbone, respectively. Considering the deeper layers of the LLM backbone have already learned fused representations of visual and textual information through extensive cross-modal training, we refine the strategy by uniformly masking the output of the second-to-last transformer block, the deepest transformer layer before the final prediction head. This modification enables $\Delta$-AttnMask to assess how visual and textual representations jointly influence the model's final interpretation, while maintaining computational efficiency. As illustrated in Figure 1, we compute average self-attention scores across attention heads to identify salient tokens, then mask the top-$p\%$ fraction of hidden states corresponding to high-attention visual and textual hidden states.

## 3.4 THEORETICAL JUSTIFICATION

We further provide a theoretical proof sketch in Appendix A.2.1, where we analyze $\Delta$-AttnMask through the lens of *Effective Mutual Information* ($I_{\text{eff}}$) (Hu et al., 2025b). $I_{\text{eff}}$ is a measure of the information actively used by the model during inference. We show that prioritizing samples with high loss delta $\Delta$ corresponds to selecting data that maximizes $I_{\text{eff}}$ between inputs and predictions. Specifically, large $\Delta$ implies that the sample contains valuable and interpretable information that helps the model predict correctly, indicating high mutual information utilization. By favoring such samples, $\Delta$-AttnMask enhances the informativeness of the training distribution, leading to faster convergence and improved generalization. This theoretical perspective supports the empirical effectiveness of our method in identifying high-utility training examples, as shown in following experiments.

## 4 EXPERIMENT

### 4.1 EXPERIMENT SETUP

#### 4.1.1 MODELS AND DATASETS

To comprehensively validate $\Delta$-AttnMask in diverse and varied real-world VIF scenarios, we begin with a small model to verify its effectiveness. We select the latest VLM model from the Qwen-VL family with an open-source base model, Qwen2-VL 2B (Wang et al., 2024b), and a small dataset from MiniGPT-4 (Zhu et al., 2023). We then test a larger and more practical model, Qwen2-VL 7B. Further, we evaluate $\Delta$-AttnMask on larger datasets, including LLaVA Instruction 158K (Liu et al., 2023) and Vision Flan 191K (Xu et al., 2023). Finally, we test our method on another model family, Llama-3.2-11B-Vision (Meta, 2024), the latest open-source VLM from the Llama family, and compare it with baselines. The evaluation benchmark details are provided in Appendix A.1.

#### 4.1.2 BASELINES AND EXPERIMENT SETTINGS

We begin the comparison with the full dataset as a strong baseline, aiming to achieve equivalent or even superior performance with less data. To further demonstrate the effectiveness of $\Delta$-AttnMask, we also include an additional comparison with the reversed $\Delta$-AttnMask, denoted as $\nabla$-AttnMask, which selects samples with the lowest loss difference. We expect this variant to perform poorly. Next, we compare $\Delta$-AttnMask with two recent strong baselines: SELF-FILTER (Chen et al., 2024) from ACL which report best results on the LLaVA Instruction 158K and PreSel (Safaei et al., 2025) from CVPR which report best results on Vision Flan 191K. For fair comparison, we use the best data portion and settings reported in their papers, and strictly equal portions of data as selected subsets for $\Delta$-AttnMask, testing uniformly on Llama-3.2-11B-Vision. For training settings and hyperparameters, we follow the default configurations of Qwen2-VL models and Llama-3.2-11B-Vision as logged in (Zheng et al., 2024); detailed settings are provided in Appendix A.5.

### 4.2 VERFICATION EXPERIMENTS RESULTS

As shown in Table 1, we first verify $\Delta$-AttnMask across multiple model scales (Qwen2-VL 2B/7B) and datasets (MiniGPT-4, LLaVA-Instruct 158K, Vision Flan 191K). The results demonstrate consistent improvements in both efficiency and performance.

For Qwen2-VL 2B on MiniGPT-4, our method achieves a +3.3% higher average score (0.462 to 0.495) using only 20% of data, with notable gains in factual accuracy (+4.6%) and MMBench performance (+4.3%). The improvements scale with model size - Qwen2-VL 7B shows a +9.7% average improvement (0.506 to 0.603), with robust gains in question accuracy (+11.6%) and MME Perception (+22.8%).

Across different datasets, $\Delta$-AttnMask maintains its effectiveness. On LLaVA-Instruct 158K, it achieves a +2.2% higher average score (0.500 to 0.522). For Vision Flan 191K, it matches the full dataset performance (0.590 vs 0.591 average) while using only 20% of the data, with additional improvements in ScienceQA (+5.4%).

Table 1: Verification experiment results for Qwen2-VL models (2B/7B) trained on different dataset configurations. Datasets: MG4 (MiniGPT-4), LLaVA (LLaVA-Instruction 158K), VFlan (Vision-Flan 191K), SQA (ScienceQA), Hallusion (HallusionBench), SEED (SEEDBench). $\Delta 20\%$ is a subset selected by $\Delta$-AttnMask. HallusionBench reports accuracy (%). MMBench, POPE, SQA, and SEEDBench report accuracy [0, 1]. MME Perception (Per) and Cognition (Cog) scores are the sum of accuracy and $accuracy+$ (Fu et al., 2024). All scores are normalized to [0, 1] for the average (Avg). Abbreviations are consistent in subsequent tables.

| Model | Config | Hallusion | | | MMBench | MME | | POPE | SQA | SEED | Avg |
|---|---|---|---|---|---|---|---|---|---|---|---|
| | | aAcc | fAcc | qAcc | | Per. | Cog. | | | | |
| 2B | MG4 Full | 43.32 | 15.90 | 14.95 | 0.53 | 1100 | 262 | 0.76 | 0.63 | 0.62 | 0.461 |
| 2B | MG4 $\Delta 20\%$ | 43.85 | 20.52 | 11.65 | 0.57 | 1231 | 268 | 0.87 | 0.65 | 0.64 | **0.495** |
| 7B | MG4 Full | 47.00 | 20.52 | 17.80 | 0.56 | 1322 | 245 | 0.88 | 0.68 | 0.62 | 0.506 |
| 7B | MG4 $\Delta 20\%$ | 57.10 | 29.77 | 29.45 | 0.67 | 1625 | 416 | 0.84 | 0.75 | 0.67 | **0.603** |
| 2B | LLaVA Full | 47.42 | 20.52 | 14.73 | 0.58 | 1158 | 300 | 0.85 | 0.65 | 0.64 | 0.500 |
| 2B | LLaVA $\Delta 20\%$ | 49.00 | 22.00 | 16.92 | 0.59 | 1203 | 375 | 0.86 | 0.64 | 0.65 | **0.522** |
| 2B | VFlan Full | 56.68 | 25.72 | 26.59 | 0.68 | 1506 | 415 | 0.87 | 0.70 | 0.70 | 0.590 |
| 2B | VFlan $\Delta 20\%$ | 53.52 | 25.72 | 24.40 | 0.71 | 1527 | 397 | 0.85 | 0.75 | 0.72 | **0.591** |

Table 2: Comparison of masking strategies. All methods select 20% subsets except Full Dataset (100%). Strategies: **Visual Masking** - masks visual encoder outputs; **Universal Masking** - random token masking in LLM backbone; $\Delta$-**AttnMask** - attention-guided masking of high-attention tokens; **Dual-Product/Sum/TOPSIS** - combine Visual Masking and $\Delta$-AttnMask scores using weighted product, weighted sum, and TOPSIS methods respectively.

| Config | Hallusion | | | MMBench | MME | | POPE | SQA | SEED | Avg |
|---|---|---|---|---|---|---|---|---|---|---|
| | aAcc | fAcc | qAcc | | Per. | Cog. | | | | |
| Full 100% | 43.32 | 15.90 | 14.95 | 0.53 | 1100 | 262 | 0.76 | 0.63 | 0.62 | 0.4614 |
| Visual 20% | 16.40 | 3.76 | 7.25 | 0.59 | 618 | 41 | 0.71 | 0.65 | 0.64 | 0.3578 |
| Universal 20% | 43.01 | 18.21 | 14.07 | 0.56 | 1259 | 216 | 0.87 | 0.62 | 0.64 | 0.4820 |
| $\Delta$-Attn 20% | 43.85 | 20.52 | 11.65 | 0.57 | 1231 | 268 | 0.87 | 0.65 | 0.64 | 0.4949 |
| Dual-Prod 20% | 44.27 | 19.65 | 16.26 | 0.56 | 1156 | 276 | 0.85 | 0.63 | 0.64 | 0.4890 |
| Dual-Sum 20% | 44.16 | 17.05 | 14.73 | 0.57 | 1223 | 215 | 0.86 | 0.65 | 0.65 | 0.4855 |
| Dual-T 20% | 47.11 | 21.39 | 18.90 | 0.57 | 1222 | 232 | 0.85 | 0.63 | 0.64 | **0.4953** |

## 4.3 $\Delta$-ATTNMASK ALTERNATION RESULTS

We evaluate various masking strategies for data selection to identify the optimal masking target, with each strategy selecting a 20% subset of the MiniGPT-4 dataset. The Non-Masking baseline uses the full dataset, while Visual Masking selects samples based on the loss delta obtained from randomly masking outputs of the visual encoder. Universal Masking applies the same framework but performs random token masking within the LLM backbone. Building upon Universal Masking and more precisely, $\Delta$-AttnMask employs attention-guided masking, selectively masking high-attention tokens in the second-to-last transformer block. Dual Masking combines the scores from Visual Masking and $\Delta$-AttnMask using multiple criteria decision analysis methods such as Weighted Product, Weighted Sum, and TOPSIS (Chakraborty, 2022).

Results in Table 2 show that $\Delta$-AttnMask achieves the second-highest average score (0.4949), outperforming all variants except Dual Masking with TOPSIS (0.4953). However, this marginal gain (+0.0004) comes at a significant computational cost increase: Dual Masking requires three inference passes per sample (baseline, visual mask, LLM backbone mask), while $\Delta$-AttnMask needs only two (baseline and masked), with a single masking operation needed, we provided a more detailed runtime analysis in Appendix A.4.

Notably, despite using random masking, the ablated variant of $\Delta$-AttnMask, Universal Masking, achieves a score of 0.4820, outperforming the full-dataset baseline. This demonstrates that the loss delta signal itself is a strong indicator of data quality when applied within the fused representation space. In contrast, Visual Masking performs poorly (0.3578), suggesting that early perturbations lead to unrecoverable information loss, preventing the LLM backbone from capturing meaningful cross-modal semantics.

Table 3: Baseline Comparison Results. Here, SF denotes SELF-FILTER, PS denotes PreSel, and $\triangledown$ represents the reversed $\Delta$-AttnMask. The best results are highlighted in bold, and the second-best results are marked with an underline.

| Config | Hallusion | | | MMBench | MME | | POPE | SQA | SEED | Avg |
|---|---|---|---|---|---|---|---|---|---|---|
| | aAcc | fAcc | qAcc | | Per. | Cog. | | | | |
| LLaVA Full | 45.43 | 17.34 | 11.43 | 0.56 | 1065 | 316 | 0.82 | 0.73 | 0.63 | 0.491 |
| LLaVA SF15.9% | 47.63 | 19.36 | 15.60 | 0.61 | 1061 | 328 | 0.72 | 0.78 | 0.60 | 0.497 |
| LLaVA $\triangledown$15.9% | 47.84 | 19.36 | 22.86 | 0.62 | 1132 | 274 | 0.81 | 0.66 | 0.66 | 0.506 |
| LLaVA $\Delta$15.9% | 49.00 | 22.25 | 16.48 | 0.67 | 1211 | 308 | 0.84 | 0.79 | 0.69 | **0.540** |
| VFlan Full | 52.37 | 21.10 | 23.74 | 0.59 | 1416 | 293 | 0.87 | 0.63 | 0.62 | **0.529** |
| VFlan PS15% | 30.07 | 9.54 | 7.03 | 0.50 | 1136 | 241 | 0.83 | 0.63 | 0.63 | 0.453 |
| VFlan $\triangledown$15% | 52.68 | 20.81 | 23.52 | 0.64 | 285 | 240 | 0.03 | 0.69 | 0.68 | 0.383 |
| VFlan $\Delta$15% | 45.43 | 13.87 | 13.63 | 0.60 | 1134 | 288 | 0.83 | 0.61 | 0.68 | 0.486 |

Table 4: Cross-Architecture Generalization Results. Configurations with "T" indicate that the $\Delta$-AttnMask subset was selected using Qwen2-VL-7B and used to train the larger Llama-3.2-11B-Vision model.

| Config | Hallusion | | | MMBench | MME | | POPE | SQA | SEED | Avg |
|---|---|---|---|---|---|---|---|---|---|---|
| | aAcc | fAcc | qAcc | | Per. | Cog. | | | | |
| LLaVA $\Delta$15.9% T | 45.95 | 19.92 | 16.09 | 0.66 | 1102 | 323 | 0.75 | 0.77 | 0.65 | 0.512 |
| VFlan $\Delta$15% T | 42.84 | 15.36 | 12.86 | 0.56 | 1013 | 273 | 0.81 | 0.60 | 0.66 | 0.466 |

We conclude that $\Delta$-AttnMask captures nearly all the benefit of more complex dual masking approaches while being simpler and more efficient. The attention-guided mechanism effectively identifies critical information, eliminating the need for multi-path evaluation or signal fusion. With only two forward passes and one masking step, $\Delta$-AttnMask provides a practical, high-performance solution for VLM data selection.

## 4.4 MAIN RESULTS

We compare $\Delta$-AttnMask against strong baselines using both the LLaVA-Instruct-158K and Vision-Flan-191K datasets, evaluating across six benchmarks and reporting an overall average score for comprehensive comparison. All methods use Llama-3.2-11B-Vision, with subset sizes matched to the best reported configurations from prior work, i.e., 15.9% for LLaVA and 15% for VFlan.

On the LLaVA setup, as shown in Table 3, $\Delta$-AttnMask achieves an average score of **0.540**, outperforming the full-dataset baseline (0.491) and the SELF-FILTER (0.497) by a significant margin, despite using only 15.9% of the data. It shows particularly strong gains in hallucination reduction, improving Hallusion aAcc to 49.00 and qAcc to 16.48, indicating superior factual consistency and question-aware reasoning. In contrast, the reversed variant $\triangledown$-AttnMask, which selects least-informative samples, underperforms despite a slight gain over full training, confirming the importance of directional sample selection.

For VFlan, $\Delta$-AttnMask reaches an average of **0.486**, surpassing the state-of-the-art PreSel baseline (0.435). It improves performance on POPE (0.83) and ScienceQA (0.61), demonstrating better generalization and truthfulness. Notably, $\triangledown$-AttnMask collapses on POPE with a score of only 0.03, highlighting the risk of poor sample selection and further validating the design of $\Delta$-AttnMask.

Crucially, $\Delta$-AttnMask is the only method that achieves higher performance than training on the full dataset across most datasets, while using less than 20% of the data. It consistently excels in reducing hallucinations, enhancing reasoning, and maintaining robust generalization, demonstrating that attention-guided loss difference is a powerful criterion for data curation in VIF.

## 4.5 CROSS-ARCHITECTURE GENERALIZATION

To assess the architectural generalization and scalability of our method, we conducted an additional experiment where data selection is performed using a small model but applied to train a significantly larger one. Specifically, we employed Qwen2-VL-7B to compute the $\Delta$-AttnMask scores and select

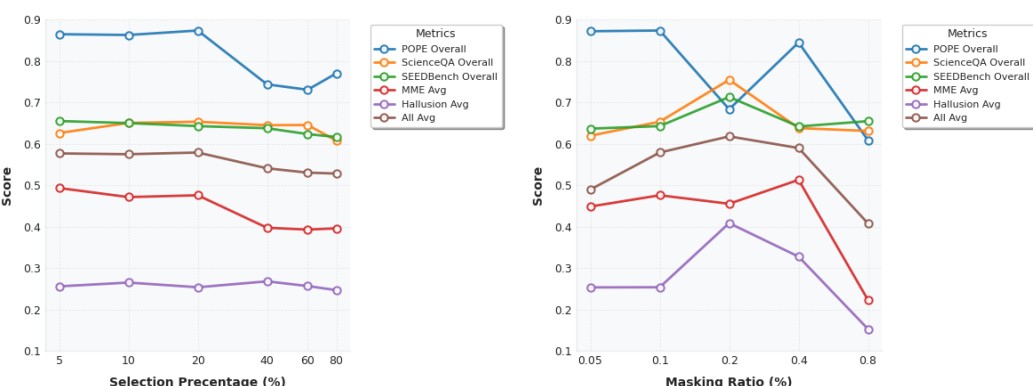

Figure 2: Ablation on selection ratio. Experiment on the MiniGPT-4 dataset with the Qwen2-VL 2B model.

Figure 3: Ablation on masking ratio. Experiment on the MiniGPT-4 dataset with the Qwen2-VL 2B model.

the optimal subsets (15.9% of LLaVA-Instruct-158K and 15% of Vision-Flan-191K). These selected subsets were then used to train a larger model, Llama-3.2-11B-Vision. The results are presented in Table 4. The configurations marked with T denote this transfer setup.

As shown in Table 4, training the larger Llama-3.2-11B-Vision model on the subsets selected by the smaller Qwen2-VL-7B yields strong performance. For the LLaVA dataset, the $\Delta$-AttnMask (T) configuration achieves an average score of 0.512, which still surpasses the full-dataset training result of the same large model (0.491). While there is a predictable drop compared to using the same 11B model for selection (0.540), the retained performance is highly competitive.

This experiment provides key insights: The $\Delta$-AttnMask criterion, derived from a small 7B model, generalizes effectively to guide the training of a larger 11B model. The fact that the selected subset yields performance superior to strong baselines and remains competitive with full-dataset training underscores that our method captures intrinsic data quality characteristics that are transferable across model architectures and scales. This demonstrates a strong cross-architecture generalization ability, emphasizing that our selection is not merely tailored to a specific model's inductive bias but identifies broadly beneficial samples. This property significantly enhances the method's practical scalability and utility, as expensive selection can be offloaded to a smaller, more efficient model before training a larger target model.

## 4.6 Ablation Experiments on Selection Ratio

We conduct an ablation study to analyze the impact of the selection ratio in $\Delta$-AttnMask on overall performance. As shown in Figure 2, the model achieves its highest average score at a selection ratio of 20%, with a performance peak of 0.4949. This indicates that sparsely attending to a small but informative subset of tokens yields optimal generalization across multiple benchmarks.

Performance remains relatively stable between 5% and 20%, suggesting that the method is effective even at very low selection ratios. The results also reveal that $\Delta$-AttnMask is moderately sensitive to this hyperparameter within the 5–20% range. Depending on the dataset's overall quality and distribution, we recommend starting with a conservative selection ratio (e.g., 5–10%) for noisier or lower-quality inputs, and gradually increasing it up to 20% to assess potential performance gains.

## 4.7 Ablation Experiments on Masking Ratio

We further investigate the sensitivity of our proposed $\Delta$-AttnMask method to the masking ratio, the proportion of hidden states masked during the forward pass for the purpose of scoring sample quality. This hyperparameter controls the degree of perturbation introduced to the model's internal representations to compute the loss difference ($\Delta$), which serves as our quality signal for data selection. It is critical to reiterate that this masking is applied only during selection; the model is subsequently trained on the chosen, original unmasked data. Figure 3 summarizes the performance

Table 5: Data augmentation experiments with $\Delta$-AttnMask on MiniGPT-4 dataset using Qwen2-VL 2B model. Strategies: **20%→40% Augment** - expand 20% subset to 40% via augmentation; **40% Selection** - directly select 40% subset; **20% 2 Epochs** - train on 20% subset for 2 epochs. All methods use $\Delta$-AttnMask for data selection.

| Strategy | Hallusion | | | MMBench | MME | | POPE | SQA | SEED | Avg |
|---|---|---|---|---|---|---|---|---|---|---|
| | aAcc | fAcc | qAcc | | Per. | Cog. | | | | |
| 20%→40% Augment | 43.428 | 18.497 | 15.824 | 0.542 | 1174.7 | 223.9 | 0.843 | 0.674 | 0.630 | 0.481 |
| 40% Selection | 44.900 | 20.231 | 15.165 | 0.553 | 1018.5 | 227.9 | 0.743 | 0.645 | 0.637 | 0.464 |
| 20% 2 Epochs | 45.216 | 21.098 | 14.505 | 0.563 | 1254.8 | 269.3 | 0.850 | 0.644 | 0.652 | 0.498 |

across multiple benchmarks when selected with varying masking ratios, ranging from 0.05 (5%) to 0.8 (80%).

Overall, the results indicate that the data selection process driven by $\Delta$-AttnMask is largely robust to a wide range of masking ratios. Metrics such as ScienceQA Overall, SEEDBench Overall, MME Avg, Hallusion Avg, and the overall average remain stable across ratios from 5% to 80%, demonstrating that the quality signal derived from the loss delta is consistent under most perturbation levels. This robustness is advantageous in practice, as it reduces the need for extensive hyperparameter tuning for the selection process.

However, the effectiveness of the scoring mechanism is challenged at two extremes. When the masking ratio is very low (e.g., 5%), the perturbation to the hidden states is too minimal. The resulting loss difference becomes negligible and less tangible, providing an insufficient signal to reliably distinguish between high- and low-quality samples. Conversely, when the masking ratio is excessively high (e.g., 80%), the hidden states are overly corrupted. This severe obstruction prevents the model from performing any meaningful forward pass, causing the loss to reflect a state of near-random guessing. In this scenario, the loss delta is no longer an informative measure of the original sample's quality, as the critical information needed for the calculation has been destroyed.

Between these uninformative extremes lies a broad plateau of effective ratios. A notable observation within this plateau is the peak in POPE Overall score when using a 10% masking ratio for selection. This suggests that a mild perturbation is optimal for identifying samples that enhance the model's robustness against hallucinations during subsequent training on the clean data. Interestingly, this same 10% ratio coincides with a slight dip for metrics like MMBench Overall and Hallusion Avg, indicating that the "best" ratio for selection might be task-dependent, likely tied to the type of noise or challenge inherent in the target benchmark.

Based on the consistent performance across most metrics, we recommend a masking ratio of 20% for general use in the selection process. This value avoids the uninformative extremes and falls squarely within the stable performance region. It introduces sufficient perturbation to generate a meaningful and discriminative loss delta for reliable quality estimation, without undermining the scoring mechanism.

Notably, all non-ablation experiments in this paper were conducted using a masking ratio of 10% for selection, a deliberately non-optimal setting for some benchmarks as revealed here. The fact that $\Delta$-AttnMask still achieves superior performance against baselines even with this suboptimal selection hyperparameter further underscores the general efficacy and robustness of our approach. It demonstrates that the core principle of using attention-guided loss delta as a quality signal is powerful, even when the implementation details are not finely tuned.

## 5 $\Delta$-ATTNMASK AS DATA AUGMENTATION

Lastly, we evaluate $\Delta$-AttnMask as a plug-in data augmentation by first selecting the top 20% of samples using the $\Delta$-AttnMask. We train the Qwen2-VL 2B on this subset for one epoch using standard forward and backward passes. In the second epoch, we reuse the exact same 20% subset but modify the forward pass by applying hidden state masking as detailed in 3.3. The rest of the network

processes the masked hidden states to produce outputs, and the loss is computed against the original target $y^*$, creating a form of targeted semantic disruption.

Crucially, because the masked tokens are those the model itself attends to most during clean inference, their removal forces the model to either recover from the loss of critical information. This induces a regularization effect: the model learns not to over-rely on any single high-attention token and instead builds more distributed, robust representations. Moreover, since the masking is only applied to already high-quality samples, those where attention is likely meaningful. Thus, the perturbations remain semantically coherent and informative, avoiding the noise injection typical of random augmentation.

As shown in Table 5, this two-phase training is denoted $\Delta$-AttnMask 20% → 40%. It uses only 20% of the full dataset but effectively doubles training exposure on the most informative samples, now augmented with model-guided perturbations.

Results show that $\Delta$-AttnMask 20% → 40% achieves an average score of 0.4815 across nine benchmarks, significantly outperforming $\Delta$-AttnMask 40% (0.4639) despite using half the number of unique samples. It also reduces hallucination, scoring 0.8432 on POPE versus 0.7431 for the 40% baseline, indicating stronger grounding. Compared to training the best 20% for two full epochs (0.4979 average), our method reaches 96.7% of that performance without seeing any new data in the second pass.

The results demonstrate that $\Delta$-AttnMask is not only effective for data selection but also serves as a seamless training-time augmentation. Perturbing high-attention regions in high-quality samples introduces meaningful semantic noise that improves robustness and generalization. This plug-in capability allows it to be integrated into standard training pipelines to enhance data efficiency and model performance without architectural changes or additional data collection.

## 6    LIMITATIONS AND FUTURE DIRECTIONS

While $\Delta$-AttnMask offers a lightweight and effective approach to data selection and augmentation for VLMs, it has certain limitations. Our method assumes the availability of meaningful attention maps from transformer-based architectures, which may restrict its direct application to non-transformer models. Finally, $\Delta$-AttnMask currently operates during post-training; adapting it for continual or online learning scenarios remains unexplored.

Future work could extend $\Delta$-AttnMask in several directions. One promising avenue is to generalize the masking mechanism to other model families beyond transformers. Another is to develop adaptive masking strategies that dynamically adjust the masking ratio based on dataset characteristics. The data augmentation capability of $\Delta$-AttnMask also opens rich research opportunities. For example, the method could be extended to support multi-step augmentation, combined with contrastive or adversarial training, or adapted for generating synthetic multimodal examples. $\Delta$-AttnMask could also be integrated into curriculum learning frameworks or used to guide synthetic data generation. Finally, applying the core idea of using model-internal sensitivity as a quality signal to other multimodal tasks and modalities (e.g., audio-video) represents a broad direction for future research.

## 7    CONCLUSION

In this work, we introduce $\Delta$-AttnMask, a principled and scalable method for data selection in VLMs that leverages the model's own sensitivity to attention-guided perturbations as a proxy for sample quality. We provide a rigorous theoretical foundation showing that $\Delta_i$ correlates with true sample quality under realistic assumptions, establishing $\Delta$-AttnMask as a theoretically grounded alternative to heuristic or model-agnostic filtering. Beyond selection, $\Delta$-AttnMask naturally extends to a plug-in data augmentation module: reusing the top-$p\%$ high-quality samples with on-the-fly hidden state masking significantly boosts generalization while reducing hallucinations. Extensive experiments across diverse vision-language benchmarks show that $\Delta$-AttnMask enables strong performance with fewer, better-curated samples. The method is lightweight, requires no additional annotations or auxiliary models. Together, these results position $\Delta$-AttnMask not only as an effective data selection tool but as a unified framework for quality-aware, self-guided multimodal learning, bridging the gap between data efficiency and scalable training for the community.

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

# A APPENDIX

## A.1 EVALUATION BENCHMARKS

To evaluate $\Delta$-AttnMask, we utilized six benchmarks: HallusionBench (Guan et al., 2024) tests image-context reasoning for language hallucination and visual illusions; MMBench (Liu et al., 2024) assesses multimodal capabilities with a bilingual dataset; MME (Fu et al., 2024) evaluates perception and cognition across 14 subtasks; POPE (Li et al., 2023b) measures object hallucination in VLMs; ScienceQA (Lu et al., 2022) tests scientific reasoning with 21,208 multimodal questions; and SEEDBench (Li et al., 2023a) evaluates hierarchical multimodal capabilities with 19,000 questions. These benchmarks collectively provide a robust and multifaceted evaluation framework, enabling us to thoroughly assess $\Delta$-AttnMask's performance across diverse tasks and domains.

## A.2 THEORETICAL ANALYSIS

### A.2.1 THEORETICAL FOUNDATION

This section establishes the mathematical framework necessary for analyzing the relationship between model performance, robustness, and data alignment in vision-language models. Our objective is to demonstrate that the $\Delta$-score, which measures sensitivity to attention masking, serves as a reliable indicator of data quality by reflecting the underlying minimum achievable loss and effective information content.

### A.2.2 MINIMUM ACHIEVABLE LOSS

For a model $f_{\boldsymbol{\theta}}$ parameterized by $\boldsymbol{\theta}$, the minimum achievable cross-entropy loss on a dataset $\mathcal{D}$ equals the conditional entropy of labels $y$ given inputs $(x^v, x^t)$:

$$\min_{\boldsymbol{\theta}} \mathcal{L}_{\text{CE}}(x^v, x^t; \boldsymbol{\theta}) = H(y \mid x^v, x^t), \tag{1}$$

where $H(y \mid x^v, x^t)$ quantifies the uncertainty in $y$ conditioned on the inputs. For well-aligned samples where $y$ is a deterministic function of $(x^v, x^t)$, we have $H(y \mid x^v, x^t) = 0$, resulting in a minimum loss of zero. Conversely, corrupted samples with $H(y \mid x^v, x^t) > 0$ necessarily incur a strictly positive minimum loss (Hu et al. 2025). This fundamental relationship establishes conditional entropy as the theoretical lower bound for cross-entropy loss, providing a principled measure of data quality.

### A.2.3 MUTUAL INFORMATION

Mutual information $I(X; Y)$ quantifies the statistical dependence between random variables $X$ and $Y$ through the relationship:

$$I(X; Y) = H(X) - H(X \mid Y), \tag{2}$$

where $H(\cdot)$ denotes Shannon entropy. In our context, mutual information between inputs $(x^v, x^t)$ and labels $y$ reveals how much information the inputs provide about the expected outputs. This quantity is essential for understanding the information-theoretic limits of model performance (Ent 2001; Shannon 1948)

### A.2.4 EFFECTIVE MUTUAL INFORMATION ($I_{\text{EFF}}$)

The effective mutual information $I_{\text{eff}}$ extends standard mutual information by accounting for model-dependent limitations in information utilization (Hu et al. 2025):

$$I_{\text{eff}}(x^v, x^t; y \mid \boldsymbol{\theta}) = I(x^v, x^t; y) - \bar{\epsilon}_{\boldsymbol{\theta}}, \tag{3}$$

where $I(x^v, x^t; y) = H(y) - H(y \mid x^v, x^t)$ represents the standard mutual information, and $\bar{\epsilon}_{\boldsymbol{\theta}}$ captures irreducible errors due to model architecture constraints or approximation noise. By combining equations equation 1 and equation 3, the minimum achievable loss can be equivalently expressed as:

$$\min_{\boldsymbol{\theta}} \mathcal{L}_{\text{CE}}(x^v, x^t; \boldsymbol{\theta}) = H(y) - I_{\text{eff}}(x^v, x^t; y \mid \boldsymbol{\theta}). \tag{4}$$

This formulation directly connects information-theoretic quantities to practical model performance, demonstrating that higher effective information corresponds to lower achievable loss (Hu et al. 2025).

### A.2.5 PROBLEM FORMULATION

Consider a vision-language model $M$ parameterized by $\boldsymbol{\theta}$ that maps visual input $x^v \in \mathcal{X}^v$ and textual input $x^t \in \mathcal{X}^t$ to a distribution over responses $y \in \mathcal{Y}$. The model computes the conditional likelihood $p_{\boldsymbol{\theta}}(y \mid x^v, x^t)$, with cross-entropy loss for sample $(x^v, x^t, y^*)$ given by:

$$\mathcal{L}(x^v, x^t; \boldsymbol{\theta}) = -\log p_{\boldsymbol{\theta}}(y^* \mid x^v, x^t).$$

We distinguish between two data distributions: $\mathcal{D}_{\text{good}}$ containing high-quality, well-aligned samples where $y$ is a deterministic function of $(x^v, x^t)$, and $\mathcal{D}_{\text{corrupt}}$ containing corrupted samples where $y$ exhibits stochastic dependence on the inputs due to noise or ambiguity. This distinction is formally characterized by conditional entropy:

$$H(Y \mid X^v, X^t; \mathcal{D}_{\text{good}}) = 0,$$
$$H(Y \mid X^v, X^t; \mathcal{D}_{\text{corrupt}}) = \delta > 0.$$

The minimum achievable cross-entropy loss for a model class parameterized by $\boldsymbol{\theta}$ on distribution $\mathcal{D}$ equals the conditional entropy:

$$\min_{\boldsymbol{\theta}} \mathcal{L}_{\text{CE}}(\mathcal{D}) = H(Y \mid X^v, X^t; \mathcal{D}).$$

Consequently, $\min_{\boldsymbol{\theta}} \mathcal{L}_{\text{CE}}(\mathcal{D}_{\text{good}}) < \min_{\boldsymbol{\theta}} \mathcal{L}_{\text{CE}}(\mathcal{D}_{\text{corrupt}})$ since $0 < \delta$.

To probe the model's reliance on attention mechanisms, we define the $\Delta$-AttnMask perturbation. Let $h_\ell(x^v, x^t)$ denote the hidden representation at layer $\ell$, and $A(x^v, x^t) \in \mathbb{R}^{k \times k}$ represent the average self-attention matrix across transformer blocks. The attention importance of token $j$ is quantified by $a_j = \sum_{m=1}^{k} A_{j,m}$. For fraction $p \in (0, 1)$, let $\mathcal{M}_p$ contain indices of the top-$p$ fraction of tokens ranked by $a_j$. The $\Delta$-AttnMask operator applies masking at layer $\ell^*$ by zeroing out hidden states at positions in $\mathcal{M}_p$:

$$\tilde{h}_{\ell^*} = \text{Mask}\left(h_{\ell^*}(x^v, x^t), \mathcal{M}_p\right).$$

The perturbed model output yields a conditional distribution $p_{\boldsymbol{\theta}}^{(\text{pert})}(y \mid x^v, x^t)$ and masked loss:

$$\mathcal{L}^{\text{masked}}(x^v, x^t; \boldsymbol{\theta}) = -\log p_{\boldsymbol{\theta}}^{(\text{pert})}(y^* \mid x^v, x^t).$$

The $\Delta$-score for sample $(x^v, x^t, y^*)$ measures the loss increase due to masking:

$$\Delta = \mathcal{L}^{\text{masked}}(x^v, x^t; \boldsymbol{\theta}) - \mathcal{L}(x^v, x^t; \boldsymbol{\theta}).$$

Our objective is to establish that higher expected $\Delta$-scores over $\mathcal{D}_{\text{good}}$ compared to $\mathcal{D}_{\text{corrupt}}$ reflect the lower minimum achievable loss and higher effective information of well-aligned data.

### A.2.6 PROOF SKETCH

We assume the model $M$ is trained to near-optimal performance, where empirical loss $\mathcal{L}(x^v, x^t; \boldsymbol{\theta})$ approximates the conditional entropy $H(Y \mid X^v, X^t; \mathcal{D})$ with diminishing error as optimization progresses. Under this assumption, the $\Delta$-score relates to information-theoretic quantities:

$$\Delta \approx H^{\text{masked}}(Y \mid X^v, X^t; \mathcal{D}) - H(Y \mid X^v, X^t; \mathcal{D}),$$

where $H^{\text{masked}}(Y \mid X^v, X^t; \mathcal{D})$ represents conditional entropy under the masked representation. Taking expectations over distribution $\mathcal{D}$ yields:

$$\mathbb{E}_{(x^v, x^t, y^*) \sim \mathcal{D}}[\Delta] = \mathbb{E}_{\mathcal{D}}\left[H^{\text{masked}}(Y \mid X^v, X^t)\right]$$
$$- H(Y \mid X^v, X^t; \mathcal{D}).$$

For $\mathcal{D}_{\text{good}}$ with $H(Y \mid X^v, X^t; \mathcal{D}_{\text{good}}) = 0$:

$$\mathbb{E}_{\mathcal{D}_{\text{good}}}[\Delta] = \mathbb{E}_{\mathcal{D}_{\text{good}}}\left[H^{\text{masked}}(Y \mid X^v, X^t)\right].$$

For $\mathcal{D}_{\text{corrupt}}$ with $H(Y \mid X^v, X^t; \mathcal{D}_{\text{corrupt}}) = \delta > 0$:

$$\mathbb{E}_{\mathcal{D}_{\text{corrupt}}}[\Delta] = \mathbb{E}_{\mathcal{D}_{\text{corrupt}}}\left[H^{\text{masked}}(Y \mid X^v, X^t)\right] - \delta.$$

The critical observation concerns $H^{\text{masked}}(Y \mid X^v, X^t)$ under both distributions. For high-quality samples in $\mathcal{D}_{\text{good}}$, models achieve zero uncertainty by concentrating attention on semantically critical tokens. Disrupting these tokens via $\Delta$-AttnMask causes substantial performance degradation, resulting in $\mathbb{E}_{\mathcal{D}_{\text{good}}}\left[H^{\text{masked}}(Y \mid X^v, X^t)\right] \gg 0$. In contrast, for corrupted samples in $\mathcal{D}_{\text{corrupt}}$, models already operate under inherent uncertainty $\delta$, often relying on diffuse attention patterns. Consequently, masking high-attention tokens produces a smaller relative uncertainty increase, yielding $\mathbb{E}_{\mathcal{D}_{\text{corrupt}}}\left[H^{\text{masked}}(Y \mid X^v, X^t)\right] \ll \mathbb{E}_{\mathcal{D}_{\text{good}}}\left[H^{\text{masked}}(Y \mid X^v, X^t)\right]$.

Given that $\delta > 0$ and the masked uncertainty for good data significantly exceeds that for corrupted data, we conclude:

$$\mathbb{E}_{\mathcal{D}_{\text{good}}}[\Delta] > \mathbb{E}_{\mathcal{D}_{\text{corrupt}}}[\Delta].$$

This demonstrates that samples from distributions with lower minimum achievable loss exhibit higher $\Delta$-scores on average, establishing the $\Delta$-score as a theoretically grounded indicator of data quality and model alignment. To further support our theoretical prediction, we also provided an additional verification experiment in Appendix A.3 as an direct evidence.

## A.3 Empirical Validation of Theoretical Predictions

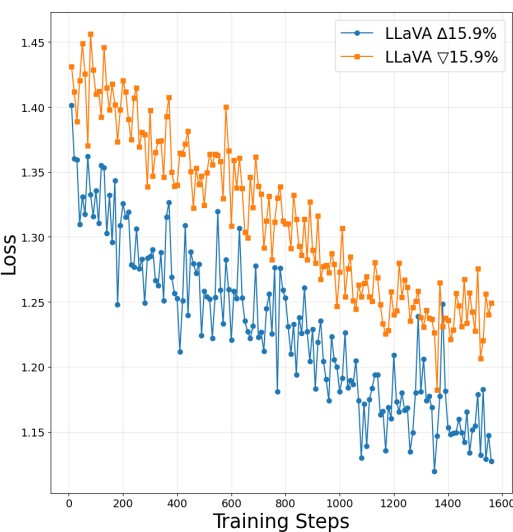

Figure 4: Training Loss Curves for Llama-3.2- 11B-Vision on LLava Instructions 158K. The x-axis denotes training steps, and the y-axis shows the cross-entropy loss. The results demonstrate that models trained on high-quality data achieve lower final losses and exhibit smoother convergence compared to those trained on corrupted data, validating the theoretical link between data alignment and minimum achievable loss

The figure 4 illustrates the training loss curves for Llama-3.2- 11B-Vision when trained on two distinct subsets of the LLava Instructions 158K dataset: one consisting of high-quality samples ($\Delta$-score $= 15.9\%$, represented in blue) and the other consisting of corrupted or misaligned samples ($\nabla$-score $= 15.9\%$, represented in orange). The results provide empirical evidence supporting the theoretical framework outlined in our paper. The model trained on the high-quality subset achieves a lower final loss compared to the model trained on the corrupted subset, which aligns with our theoretical prediction that well-aligned data leads to a lower minimum achievable loss. This is consistent with the conditional entropy formulation where $H(Y \mid X^v, X^t; \mathcal{D}_{\text{good}}) = 0$ indicates perfect alignment, while $H(Y \mid X^v, X^t; \mathcal{D}_{\text{corrupt}}) = \delta > 0$ reflects uncertainty due to misalignment. Furthermore, the training trajectory of the high-quality model exhibits smoother convergence and more consistent optimization progress, indicating a more stable learning process. In contrast, the model trained on corrupted data shows higher variance and a slower decline in loss, suggesting that noisy or misaligned inputs introduce optimization challenges and degrade the signal-to-noise ratio during training. The persistent performance gap between the two curves throughout the entire training phase underscores the critical role of data quality in determining the ultimate performance

of vision-language models. Even after extensive training, the model exposed to corrupted data fails to close the gap, indicating that data quality imposes a fundamental limit on learnability. These findings validate the core hypothesis of our work: data alignment directly influences the minimum achievable loss, with well-aligned datasets enabling models to exploit deterministic input-output relationships more effectively. The observed differences in convergence behavior further emphasize the practical importance of curating high-quality, well-aligned datasets in vision-language modeling, as they facilitate more robust, efficient, and effective training dynamics.

## A.4 RUNTIME ANALYSIS

To evaluate the computational efficiency of $\Delta$-AttnMask, we analyze its runtime in terms of floating-point operations (FLOPS) required for the data selection phase on a fully labeled dataset. We denote the dataset size as $N$ and the FLOPS for a single forward pass through the target VLM as $F$. Minor operations, such as attention averaging, masking application, or score sorting, are negligible compared to $F$ and are thus omitted.

$\Delta$-AttnMask operates through two primary variants, both leveraging inference-only passes without gradient computations or parameter updates. The dual-masking variant separately masks visual and textual hidden states, requiring three forward passes per sample: one for baseline loss computation, one for visual masking in the visual encoder, and one for textual masking in the LLM backbone. This results in approximately $3FN$ FLOPS.

To enhance efficiency, the simplified variant unifies masking at the second-to-last transformer block in the LLM backbone, where visual and textual representations are already fused. As illustrated in the method overview, this requires only two forward passes per sample: one for baseline loss and attention extraction, and one for masked loss recomputation. The attention scores are computed by averaging self-attention across heads, identifying salient tokens for masking the top-$p\%$ fraction of hidden states. This unified approach reduces FLOPS to approximately $2FN$, achieving a 33% reduction compared to the dual variant while maintaining near-equivalent performance.

Comparing with baselines, SELF-FILTER for instance, it employs the VLM itself as a filter through a two-stage process. In Stage 1, a lightweight scoring network is co-trained with the VLM over the dataset, using reweighted losses to learn sample difficulties. This involves one epoch of training, typically incurring about $3FN$ FLOPS: one forward pass plus backward passes (approximately twice the forward FLOPS in transformers) (Kaplan et al., 2020). Feature extraction via lighter models (e.g., CLIP or GPT-4 Vision) adds a smaller overhead of $CN$ FLOPS, where $C \ll F$. Stage 2 uses the trained network for scoring and applies diversity penalties via $k$-nearest neighbors, but these are secondary to the training cost.

Comparing the methods, the simplified $\Delta$-AttnMask demands $2FN$ FLOPS, approximately $\frac{2}{3}$ that of SELF-FILTER's $3FN$ FLOPS (excluding minor terms). This efficiency advantage stems from $\Delta$-AttnMask's inference-only design, avoiding the overhead of gradient-based optimization and co-training. Even the dual variant matches SELF-FILTER's cost but offers flexibility for scenarios prioritizing marginal performance gains. Overall, $\Delta$-AttnMask provides a more scalable solution for large datasets, where reducing passes directly translates to practical runtime savings without sacrificing effectiveness.

## A.5 TRAINING SETTINGS

- Gradient Accumulation Steps: 2
- Per Device Train Batch Size: 1
- Lr scheduler type: cosin
- num training epochs: 1
- Freeze vision tower: true
- Freeze Multi Modal Projector: true
- train mm proj only: false
- Learning rate: 1e-5
- Every model is trained on 8 NVIDIA A800 GPUs

A.6 ABLATION ON MASKING LAYERS

Table 6: Ablation study on masking layers. All methods select 20% subsets of the MiniGPT-4 dataset. "Visual" masks visual encoder outputs; "Median" masks the 9th transformer block; "Deeper" masks the 18th block; "Universal" masks the second-to-last (26th) transformer block. All experiments use Qwen2-VL 2B.

| Config | Hallusion | | | MMBench | MME | | POPE | SQA | SEED | Avg |
|--------|-----------|------|------|---------|------|------|------|-----|------|-----|
| | aAcc | fAcc | qAcc | | Per. | Cog. | | | | |
| Full 100% | 43.32 | 15.90 | 14.95 | 0.53 | 1100 | 262 | 0.76 | 0.63 | 0.62 | 0.4614 |
| Visual 20% | 16.40 | 3.76 | 7.25 | 0.59 | 618 | 41 | 0.71 | 0.65 | 0.64 | 0.3578 |
| Median 20% | 19.13 | 5.21 | 12.62 | 0.60 | 693 | 48 | 0.72 | 0.66 | 0.63 | 0.3760 |
| Deeper 20% | 30.51 | 7.54 | 11.65 | 0.54 | 998 | 270 | 0.71 | 0.61 | 0.62 | 0.4242 |
| Universal 20% | 43.01 | 18.21 | 14.07 | 0.56 | 1259 | 216 | 0.87 | 0.62 | 0.64 | **0.4820** |

To investigate the impact of masking at different layers of the transformer backbone, we conducted an ablation study comparing several masking strategies, all applied to 20% of the MiniGPT-4 dataset.

As shown in Table 6, Universal Masking achieves the highest average score (0.4820), outperforming all other layer-specific strategies. This indicates that masking within the fused representation space of the LLM backbone provides a strong signal for data quality assessment.

In contrast, Visual Masking performs poorly (0.3578), with significant degradation in hallucination metrics; for instance, the `aAcc` score drops from 43.32 to 16.40. This suggests that perturbing visual features before cross-modal integration leads to irreversible information loss, preventing the model from leveraging meaningful visual-textual interactions.

Median Masking (9th block) and Deeper Masking (18th block) show moderate performance (0.3760 and 0.4242, respectively). The improvement from Median to Deeper masking indicates that masking closer to the output captures more semantically meaningful representations, as evidenced by the higher MME Perception score (998 vs. 693).

The superior performance of masking at the second-to-last transformer block can be attributed to several key factors:

- **Sufficient Cross-Modal Fusion:** By the second-to-last layer, visual and textual representations have undergone extensive interaction through multiple transformer blocks. The hidden states at this stage encode rich, fused multimodal semantics, allowing perturbations to reflect how integrated information contributes to the final prediction. In contrast, masking earlier layers disrupts the fusion process prematurely, leading to noisy or uninterpretable loss signals.

- **Proximity to Prediction Without Interfering with Autoregressive Generation:** Masking the final layer's hidden states is ineffective because autoregressive generation relies solely on the last hidden state to predict the next token. Perturbing the final layer would directly corrupt the output distribution without capturing the model's internal reasoning. The second-to-last layer, however, provides a high-level, contextualized representation that strongly influencesthe final output while does not directly constitute, making it a sensitive yet stable target for quality assessment.

- **Semantic Richness and Stability:** Deep transformer layers capture abstract, task-relevant features. The second-to-last layer retains nuanced semantic information while being less susceptible to low-level noise compared to earlier layers. This balance ensures that the loss delta ($\Delta_i$) reflects meaningful changes in the model's understanding rather than superficial perturbations.

- **Consistency with Theoretical Motivation:** As discussed in Section 3.4 and Appendix A.2.1, $\Delta$-AttnMask aims to maximize the effective mutual information ($I_{\text{eff}}$) between inputs and predictions. The second-to-last layer's representations are both highly informative and directly predictive, ensuring that masking-induced loss differences correlate strongly with sample utility.

