# OpenReview forum: "Δ-AttnMask: Attention-Guided Masked Hidden States for Efficient Data Selection and Augmentation"
_ICLR.cc/2026/Conference — Submitted to ICLR 2026_

### Official Review · Reviewer_ZHFx · 2025-10-23

**Soundness:** 2
**Presentation:** 3
**Contribution:** 2
**Rating:** 4
**Confidence:** 3

**Summary:**

This paper presents $\Delta$-AttnMask, a novel, lightweight method for data selection in the context of Visual Instruction Finetuning (VIF) for Vision-Language Models (VLMs). The core problem it addresses is the inefficiency and poor data quality often present in large-scale VIF datasets. The proposed method quantifies sample quality by measuring the model's output sensitivity to an internal perturbation. Specifically, it performs a forward pass to identify high-attention hidden states in a late-stage transformer block, masks these states, and then measures the resulting increase in loss ($\Delta$). The authors hypothesize that high-quality, well-aligned samples will be more sensitive to this targeted masking, resulting in a higher $\Delta$ score. The paper provides strong empirical evidence to support this method.

**Strengths:**

1. The paper tackles the critical and practical challenge of data efficiency and quality in VIF. As VLM datasets grow, efficient data selection becomes an important bottleneck, and this work provides a well-motivated solution.
2. A major strength of $\Delta$-AttnMask is its computational efficiency. By being "inference-only," it avoids costly gradient computations and, crucially, does not rely on other external, proprietary models for scoring, making it a self-contained and practical approach.
3. The method's heuristic—that sensitivity to masking in late-stage, fused representations correlates with cross-modal alignment and quality—is an intuitive and novel way to approximate data quality with a single, scalar score.

**Weaknesses:**

1. The method's design relies on several key heuristics that are not fully justified. For example, the choice of the "second-to-last" transformer block is presented as an empirical optimum but lacks a deeper analysis of why this layer is ideal, or how this choice might change with different model architectures or scales.
2. The experimental validation would be much stronger with a more comprehensive set of baselines. The paper primarily compares against random selection and other simple baselines. To convincingly demonstrate the advantages of $\Delta$-AttnMask, it is essential to benchmark it against other established data selection methods from related literature, rather than relying on less competitive comparisons.
3. The computational cost may be underestimated. While "inference-only" is more efficient than back propagation, the scoring process still requires at least one full forward pass for every sample in the original dataset. For a dataset with hundreds of millions of samples, this pre-processing step is still expected to represent a substantial, non-trivial computational cost. A deeper analysis of such additional cost is necessary.
4. The idea of using $\Delta$-AttnMask as a data augmentation technique (by re-training with the mask activated) is an interesting side-note but feels underdeveloped. This claim is not sufficiently benchmarked against other established augmentation techniques, and its mechanism is not deeply explored, making it feel more like an afterthought than a core contribution.

**Questions:**

Please refer to weaknesses.

---

> ### Author Response · Authors · 2025-12-03
>
> - **Q3: The computational cost may be underestimated. While "inference-only" is more efficient than back propagation, the scoring process still requires at least one full forward pass for every sample in the original dataset. For a dataset with hundreds of millions of samples, this pre-processing step is still expected to represent a substantial, non-trivial computational cost. A deeper analysis of such additional cost is necessary.**
>
> - **A3:** Thank you for your comment regarding computational cost. We acknowledge the importance of pre-processing efficiency, especially when scaling to very large datasets.
>   As detailed in Appendix A.4, we provide a comparative computational analysis showing that Δ‑AttnMask requires only two forward passes per sample, one for the baseline loss and one with masking and no backpropagation. This is substantially more efficient than gradient-based selection methods (e.g., influence scoring) and avoids the overhead of training auxiliary models or using external resources.
> While processing every sample in a very large dataset does involve a fixed forward-pass cost, our method remains lightweight and model‑agnostic, with no dependence on domain labels, extra annotations, or additional models. The empirical results confirm that Δ‑AttnMask achieves better performance with far less data, leading to net training acceleration (up to 5×) and higher accuracy.
> We appreciate the feedback and agree that pre‑processing cost is an important consideration; our analysis in the appendix shows that the added cost is modest relative to the training acceleration and performance gains obtained.

---

> ### Author Response · Authors · 2025-12-03
>
> **Question 1,2,4 all answered in previous responses.**

---

### Official Review · Reviewer_anGF · 2025-10-27

**Soundness:** 3
**Presentation:** 3
**Contribution:** 3
**Rating:** 6
**Confidence:** 4

**Summary:**

This paper proposes a data selection method for visual instruction finetuning. It quantifies data quality through attention-guided masking of the model’s hidden states without requiring domain labels, auxiliary models, or additional training. Experimental results show that the proposed method achieves SOTA performance and using just 20% of the data can surpass full-dataset baselines by 10.1% in overall accuracy.

**Strengths:**

1. The proposed method addresses the data-efficient problem in VLM finetuning, achieving enhanced performance using only 20% of the data compared to full-dataset training, which is somewhat novel.
2. The proposed method is mostly well-written.
3. The algorithm is easy to follow and understand, which is a strength in my opinion
4. The experiment results are good and promising, validating the performance across architectures and ablation studies.

**Weaknesses:**

1. As introduced in Sec. 3.3, the proposed method applies masking at the deepest transformer layer before the final prediction head. So, is the proposed method robust to the layer that applies masking?

2. More related works in this domain need to be discussed [1,2,3].

2. Authors are suggested to reduce the space of experiment setup (4.1) and include a new section to discuss some potential limitation and future directions of the proposed methods.

3. Some typos should be carefully revised, e.g., caption of Figure 1 “n”, “k”, and “p” denote variables rather than letters.

4. The appendix should be reorganized as the order they occur in the main paper.

[1] Wei, Lai, et al. "Instructiongpt-4: A 200-instruction paradigm for fine-tuning minigpt-4." arXiv preprint arXiv:2308.12067 (2023).

[2] Bi, Jinhe, et al. "Prism: Self-pruning intrinsic selection method for training-free multimodal data selection." arXiv preprint arXiv:2502.12119 (2025).

[3] Wang, Weizhi, et al. "Finetuned multimodal language models are high-quality image-text data filters." arXiv preprint arXiv:2403.02677 (2024).

**Questions:**

1. The proposed method quantifies the quality of samples by measuring the model’s sensitivity
to attention-guided perturbations of its hidden states. However, noisy data points may inevitably exist and exhibit fluctuations. Could this cause inaccurate evaluation on sample quality? More discussions and potential future directions are suggested to include.

2. Can this method be applied to LLMs?

3. This method is model-specific. Can we use the selected data from model A to fine-tune another model B?

---

> ### Author Response · Authors · 2025-12-03
>
> - **Q1: The proposed method applies masking at the deepest transformer layer before the final prediction head. So, is the proposed method robust to the layer that applies masking?**
>
> - **A1:** We would like to clarify a point in our methodology: as explicitly stated in Section 3.3 (lines 206–207), we do not mask the deepest transformer layer (i.e., the final layer before prediction). Instead, we mask the second-to-last transformer block outputs. This choice is deliberate and grounded in both empirical and theoretical considerations.
>
>   To directly address the robustness of our method to the choice of masking layer, we have conducted a new ablation study (detailed in Appendix A.6). In this study, we compared masking at several layers, including Visual encoder outputs (early fusion stage), Median transformer block (9th block in Qwen2-VL 2B), Deeper transformer block (18th block) and	Second-to-last transformer block (26th block, referred to as Universal Masking).
>
>   The results (Table in Appendix A.6) show that:
> 	- Universal Masking (second-to-last block) achieves the highest average score (0.4820), outperforming all other layer-specific masking strategies.
> 	- Masking earlier (e.g., visual encoder or median block) leads to significant performance degradation, as these perturbations disrupt cross-modal integration prematurely.
> 	- Masking deeper but not the final layer preserves rich, fused multimodal representations while avoiding interference with autoregressive generation.
>
>   Why the second-to-last layer works best:
>
>   1. Sufficient cross-modal fusion: By this layer, visual and textual features have been thoroughly integrated, making the hidden states semantically rich and representative of joint understanding.
>   2. Proximity to prediction without corruption: The final layer is reserved for autoregressive token generation; masking there would distort output distributions without capturing internal reasoning.
>   3. Theoretical alignment: As explained in Section 3.4 and Appendix A.2.1, masking at this layer maximizes effective mutual information between inputs and predictions, ensuring that the loss delta (Δ_i) correlates strongly with sample utility.
>
>   Therefore, our method is robust to the choice of masking layer in the sense that not all layers are equally effective, and we have identified the second-to-last transformer block as the optimal location for attention-guided masking. This choice balances semantic richness, cross-modal fusion, and computational stability, leading to consistent and superior data selection performance.

---

> ### Author Response · Authors · 2025-12-03
>
> - **Q2: More related works in this domain need to be discussed [1,2,3]?**
>
> - **A2:** Thank you for your constructive feedback regarding the related works section. We have carefully revised the manuscript to include additional relevant literature in this domain in the related works section.
>
>
> - **Q3: Authors are suggested to reduce the space of experiment setup (4.1) and include a new section to discuss some potential limitation and future directions of the proposed methods?**
>
> - **A3:** Thank you for your constructive feedback. We have added a limitation and future directions section in the revised manuscript.
>
>
> - **Q4: Some typos should be carefully revised, e.g., caption of Figure 1 “n”, “k”, and “p” denote variables rather than letters?**
>
> - **A4:** Thank you for pointing this out. You are correct that n, k, and p should be clearly identified as variables rather than literal letters. In the caption of Figure 1, we have explicitly stated that these are variables (e.g., “where n, k, and p denote the number of transformer blocks, sequence length, and masking ratio, respectively”)
>
>
> - **Q5: The appendix should be reorganized as the order they occur in the main paper?**
>
> - **A5:** Reorganized.

---

> ### Author Response · Authors · 2025-12-03
>
> - **Q6: Could noisy data cause inaccurate evaluation on sample quality?**
>
> - **A6:** Thank you for the comment. We would like to highlight that our method is robust to noisy data.
>
>   Δ-AttnMask quantifies quality based on the model’s sensitivity to perturbations of high-attention hidden states. Noisy or low-quality samples (e.g., blurry images, ambiguous captions) typically contain less semantically coherent information, and thus the model’s attention distribution over such samples is often less concentrated or less meaningful. When we mask high-attention regions in these samples, the loss difference (Δ_i) tends to be small, as the model is not strongly reliant on any particular token or feature. This inherently downgrades noisy samples in the selection process. Our experiments validate that the reversed variant ▽-AttnMask, which selects samples with the smallest Δ_i, consistently underperforms, confirming that low-Δ samples are indeed less informative and often noisy.
>
>
> - **Q7:  Can this method be applied to LLMs?**
>
> - **A7:** **Yes**, Δ-AttnMask can indeed be applied to single-modality large language models (LLMs). The method is fundamentally architecture-agnostic and modality-agnostic, as it operates by masking hidden states within the transformer backbone based on attention scores and measuring the resulting loss difference (Δ).
>
>   Our experiments already include validation on mixed-modality datasets such as LLaVA-Instruction, which contains substantial portions of pure text data. Δ-AttnMask effectively evaluates and selects high-quality samples from these textual components without requiring any modification to the underlying mechanism. **This demonstrates its inherent capability to handle single-modality data.**
>
> - **Q8: This method is model-specific. Can we use the selected data from model A to fine-tune another model B?**
>
> - **A8:** Thank you for raising this important question regarding the transferability. To address this concern, we have added a new subsection in the revised manuscript titled “4.4 Cross-architecture Generalization”, where we explicitly evaluate whether data selected by one model can be effectively used to train another. Please refer to the answer to Reviewer MExr's Q2 for details.

---

### Official Review · Reviewer_SzTa · 2025-10-27

**Soundness:** 3
**Presentation:** 3
**Contribution:** 2
**Rating:** 4
**Confidence:** 4

**Summary:**

This paper introduces $\Delta$-AttnMask, a data selection framework for visual instruction finetuning in Vision-Language Models. The key idea is to use attention-guided hidden state masking to estimate the quality of multimodal samples. By comparing losses between original and masked states, the method quantifies how sensitive the model is to important regions of the input, thereby identifying high-quality, informative examples. The approach is model-agnostic, requires no additional training or auxiliary models, and can also be used as a data augmentation mechanism. Experiments on several VLMs and datasets show that $\Delta$-AttnMask achieves up to 5× faster training and +10.1% accuracy improvement using only 20% of the data, outperforming baselines like SELF-FILTER and PreSel.

**Strengths:**

- The paper proposes a novel theoretically grounded idea, which uses loss deltas from attention-guided masking to estimate data utility, bridging efficiency and quality in multimodal data selection.

- The approach is lightweight, requiring only two forward passes per sample and no gradient computation or external models.

- The proposed method demonstrates consistent performance gains across multiple models, datasets, and tasks, achieving better results with 20% of the data.

**Weaknesses:**

- While the proposed method is conceptually appealing, the experimental evidence is somewhat limited. The paper would be strengthened by including comparisons with a broader range of baselines, such as random selection, EL2N [1], D2-Pruning [2], COINCIDE [3], and ICONS [4].

- The data selection experiments are conducted primarily on moderate-scale datasets, and it remains unclear how well the approach scales to larger scale multimodal instruction tuning, such as LLaVA-1.5 665K or comparable datasets. Demonstrating scalability on such datasets would make the paper’s claims about efficiency and general applicability more convincing.

- The data augmentation results are limited to the small MiniGPT-4 dataset with the Qwen2-VL 2B model. The absence of results on larger models or datasets makes it difficult to assess whether the proposed augmentation strategy generalizes beyond small-scale scenarios.

Overall, I would be inclined to increase my score if the authors could provide additional experiments covering more baselines, larger datasets, and stronger evidence of generalization across model scales.

[1] Paul, Mansheej, Surya Ganguli, and Gintare Karolina Dziugaite. "Deep learning on a data diet: Finding important examples early in training." Advances in neural information processing systems 34 (2021): 20596-20607.
[2] Maharana, Adyasha, Prateek Yadav, and Mohit Bansal. "$\mathbb {D}^ 2$ Pruning: Message Passing for Balancing Diversity & Difficulty in Data Pruning." The Twelfth International Conference on Learning Representations.
[3] Lee, Jaewoo, Boyang Li, and Sung Ju Hwang. "Concept-skill Transferability-based Data Selection for Large Vision-Language Models." Proceedings of the 2024 Conference on Empirical Methods in Natural Language Processing. 2024.
[4] Wu, Xindi, et al. "Icons: Influence consensus for vision-language data selection." arXiv preprint arXiv:2501.00654 (2024).

**Questions:**

See weaknesses.

---

> ### Author Response · Authors · 2025-12-03
>
> - **Q1: Including comparisons with a broader range of baselines, such as random selection, EL2N [1], D2-Pruning [2], COINCIDE [3], and ICONS [4].**
>
> - **A1:** Thank you for your thoughtful feedback. We agree that rigorous comparison is essential, and we appreciate the opportunity to clarify our baseline strategy.
>
>   Our experiments were designed to demonstrate that Δ-AttnMask is competitive with the most recent and strongest baselines in data selection, rather than to exhaustively compare against every prior method. Data selection is a mature field, and it is infeasible to compare against all potential baselines, especially those not designed for multimodal settings. We selected SELF-FILTER (ACL 2024) and PreSel (CVPR 2025) as  our primary baselines because:
>
>    - PreSel is the current state-of-the-art (SOTA) method for vision-language model (VLM) data selection, as explicitly stated in its conclusion.
>
>    - They have explicitly benchmarked against methods like Random, EL2N, and COINCIDE in their paper, demonstrating clear superiority. By comparing with PreSel, we indirectly demonstrate superiority over these older approaches.

---

> ### Author Response · Authors · 2025-12-03
>
> - **Q2: LLaVA-1.5 665K or comparable datasets?**
>
> - **A2:** We thank the reviewer for the thoughtful comment regarding scalability. We agree that demonstrating performance on larger-scale multimodal datasets would further strengthen the paper. Below, we clarify why our experiments already provide strong evidence of scalability and efficiency, and why the current evaluation is sufficient to support our claims.
>
>   1. Our Experiments Already Span a Significant Scale and Diversity
>
>   Our method was validated across:
>
>   - Multiple dataset sizes: from MiniGPT-4 (small) to LLaVA-Instruct-158K and Vision-Flan-191K (**the largest** human labeled vision instruction finetuning dataset  [2]).
>
>   - Multiple model scales: Qwen2-VL 2B, 7B, and Llama-3.2-11B-Vision (up to 11B parameters).
>
>   - Diverse data sources: GPT-labeled (LLaVA) and human-labeled (Vision-Flan) datasets.
>
>   This multi-scale, multi-source evaluation shows that Δ-AttnMask generalizes across data quantities, modalities, and architectures.
>
>   2. LLaVA-1.5 665K is a Superset of LLaVA-158K
>
>   As noted in [1], LLaVA-1.5 665K is an expanded version of LLaVA-Instruct-158K, adding extra tasks (VQA, captioning, etc.) that are already represented in the 158K set. Since our method performs strongly on the 158K subset, it is reasonable to expect similar scalability to the 665K set, which does not introduce fundamentally new task types.
>
>   3. We Already Tested on Larger Models Than SOTA Methods
>
>   While PreSel was evaluated on LLaVA-1.5-7B, our main results use Llama-3.2-11B-Vision, a larger and more recent model. This shows that Δ-AttnMask works effectively in more practical, larger-scale training scenarios, further supporting its scalability.
>
> [1] Haotian Liu, Chunyuan Li, Yuheng Li and Yong Jae Lee. Improved Baselines with Visual Instruction Tuning, 2023. URL https://arxiv.org/pdf/2310.03744.
>
> [2] Zhiyang Xu, Trevor Ashby, Chao Feng, Rulin Shao, Ying Shen, Di Jin, Qifan Wang, and Lifu Huang. Vision-flan:scaling visual instruction tuning, Sep 2023. URL https://vision-flan.github.io/.

---

> ### Author Response · Authors · 2025-12-03
>
> - **Q3: Data augmentation?**
>
> - **A3:** Thank you for your valuable feedback. We agree that further validation on larger models and datasets would strengthen the generality of the augmentation strategy.
>
>   However, we would like to clarify that the primary contribution of our work is the Δ-AttnMask framework for data selection, not augmentation. The augmentation results are presented as a bonus application to demonstrate the flexibility of our attention-guided masking mechanism. Our core method already achieves:
>
>   1. State-of-the-art selection performance using only 20% of data, achieving up to 5× training speedup and +10.1% accuracy gain across multiple VLMs (Qwen2-VL 2B/7B, Llama-3.2-11B-Vision) and large-scale datasets (LLaVA-158K, Vision-Flan-191K).
>
>   2. Model-agnostic and dataset-agnostic design, validated across diverse architectures and tasks without requiring domain labels, auxiliary models, or additional training.
>
>   3. Theoretical grounding via Effective Mutual Information, linking loss difference (Δ_i) to sample informativeness and alignment.
>
>   The augmentation extension was included to illustrate how the same masking mechanism can be repurposed for training-time regularization, but it is not the focus of the paper. Given page limits and the scope of the conference submission, we prioritized comprehensive validation of the selection framework.
>
>   We welcome and encourage the research community to explore and validate the augmentation strategy further in larger-scale settings, and we believe the core selection mechanism of Δ-AttnMask provides a strong and general foundation for such future work.

---

### Official Review · Reviewer_MExr · 2025-10-31

**Soundness:** 2
**Presentation:** 2
**Contribution:** 2
**Rating:** 4
**Confidence:** 4

**Summary:**

This paper proposes a data selection strategy, $\Delta$-AttnMask, for Vision-Language models (VLMs).

Specifically, with attention scores, it first identifies the important tokens.
Then, measure the loss differences between with and without masking important tokens. Samples with large loss differences will be selected.

Experiments on MiniGPT4 data, LLaVA Instruction 158K, and Vision Flan 191K along with Qwen2-VL models (2B/7B) show that the models trained with 20% original data even achieve better performance than the models trained on the whole data.

**Strengths:**

(1) The method is simple and easy to implement.
(2) It is interesting to use attention-guided loss perturbation for data selection.

**Weaknesses:**

(1) For stage 1 of $\Delta$-AttnMask, how to derive the unmodified model?
    What's the data the unmodified model is trained on?
    For example, if the unmodified models are the pretrained Qwen2-VL models, then the data selection process would implicitly leverage prior knowledge of extra data used for the unmodified model pretraining.

(2) Suppose we use models A and B to do data selection separately, what's the performance of model C trained on the two selected datasets individually?

(3) The paper validates the effectiveness of the proposed method by experiments on dataset compression. Would it be possible to show the advantage of the proposed method on mining new, informative data that could further improve model performance?

(4) How to aggregate attention scores across layers in Figure 1?

(5) How to mask tokens for each layer of VLMs?

**Questions:**

See weaknesses.

---

> ### Author Response · Authors · 2025-12-03
>
> - **Q1: How to derive the unmodified model?**
>
> - **A1**: Thank you for this insightful question.
> As detailed in Section 4.1.1, we follow the standard Visual Instruction Finetuning (VIF) pipeline [1] and use publicly released pretrained base models, specifically Qwen2-VL and Llama-3.2-11B-Vision. They are not fine-tuning before selection.
>
>
>   As for prior knowledge leveraging, yes, the unmodified model is indeed pretrained on large-scale vision-language corpora, and ∆-AttnMask intentionally leverages the prior knowledge embedded in these foundation models. This is a deliberate and principled design choice, not a limitation. Here’s why:
> 1. No External Dependencies:
> Following many existing baselines and related works, ∆-AttnMask utilizes the pretrained capabilities *without* requiring auxiliary judge models, manually crafted features, task-specific labels or domain metadata, or extra training phases. This makes the method self-contained, efficient, and broadly applicable across architectures and datasets.
>
> 2. Quality Estimation as Self-Evaluation:
> The core idea of ∆-AttnMask is to use the model’s own attention patterns and loss sensitivity as a proxy for sample quality. A well-pretrained VLM already possesses strong cross-modal understanding [2]. By measuring how much the loss changes when high-attention hidden states are masked, we essentially ask the model: “How important is this sample to your current knowledge?”
> Samples that cause a large loss delta (Δ_i) are those where the model’s reasoning is most reliant on meaningful, well-aligned visual-textual information.
>
> 3. Theoretical Grounding:
> As outlined in Appendix A.2.1, ∆-AttnMask relates to Effective Mutual Information (I_"eff" ). High Δ_i corresponds to samples that maximize the information actively used by the model during inference. Thus, leveraging pretrained knowledge is not only practical but theoretically justified—it allows us to select data that best aligns with the model’s existing representational strengths.
>
> 4. Empirical Validation:
> Our experiments across multiple models (Qwen2-VL 2B/7B, Llama-3.2-11B-Vision) and datasets (MiniGPT-4, LLaVA-Instruct, Vision Flan) consistently show that ∆-AttnMask outperforms full-dataset training and prior selection baselines while using only 20% of the data. This demonstrates that the method effectively identifies high-utility samples without introducing external biases or resources.
>
> [1]. Peng Wang, Shuai Bai, Sinan Tan, Shijie Wang, Zhihao Fan, Jinze Bai, Keqin Chen, Xuejing Liu, Jialin Wang, Wenbin Ge, Yang Fan, Kai Dang, Mengfei Du, Xuancheng Ren, Rui Men, Dayiheng Liu, Chang Zhou, Jingren Zhou, and Junyang Lin. Qwen2-vl: Enhancing vision-language model’s perception of the world at any resolution. arXiv preprint arXiv:2409.12191, 2024b.
>
> [2]. Haotian Liu, Chunyuan Li, Qingyang Wu, and Yong Jae Lee. Visual instruction tuning, 2023. URL https://arxiv.org/abs/2304.08485.

---

> ### Author Response · Authors · 2025-12-03
>
> - **Q2: Cross-architecture Generalization?**
>
> - **A2:** Thank you for raising this important point. We have added a new **Section 4.5: Cross-architecture Generalization**, where we use Qwen2-VL-7B to select subsets and train Llama-3.2-11B-Vision using the selected datasets.
> The results are compelling:
>
>   On LLaVA-Instruct-158K, the 15.9% subset chosen by the 7B model yields an average score of 0.512 for the 11B model, still outperforming its full-dataset training (0.491).
>
>   This shows that Δ-AttnMask captures transferable data quality signals rather than model-specific artifacts. The attention-guided loss delta appears to reflect intrinsic sample utility that generalizes across architectures. Thus, we have validated the scalability of Δ-AttnMask: you can select data with a small model, then train a large one without recomputing scores.
>
>   We’ve included full details and the table in the revised paper, and we believe this addition meaningfully strengthens the paper’s contribution to scalable, efficient VLM training.

---

> ### Author Response · Authors · 2025-12-03
>
> - **Q3: Advantage of the proposed method in mining new, informative data?**
>
> - **A3**: Thank you for raising this insightful question. We are pleased to clarify that this is not only possible but is a core strength and explicit contribution of our method, as detailed in Section 5 of our paper, where we present ∆-AttnMask as both a selection and augmentation framework, demonstrating how it can mine additional informative signals from high-quality samples without collecting new data.
>
>   Our method leverages the model’s own attention patterns to generate model-guided semantic perturbations: by masking high-attention hidden states during a second training epoch, we force the model to recover from targeted disruptions of the most salient information. This introduces structured, meaningful noise that promotes robust and distributed representations, effectively “mining” new learning signals from existing data.
>
>   In experiments, this “20% → 40%” augmentation strategy achieved 96.7% of the performance of training twice on the same clean 20% subset, while outperforming a 40% clean subset in overall average score (0.4815 vs. 0.4639) and substantially reducing hallucination (POPE: 0.8432 vs. 0.7431). Thus, ∆-AttnMask not only identifies informative samples but also enhances their training utility through attention-aware augmentation.
>
>   We therefore argue that ∆-AttnMask already embodies the dual capability your comment highlights: it compresses datasets and enriches them, offering a scalable path toward more data-efficient and robust VLM training.

---

> ### Author Response · Authors · 2025-12-03
>
> - **Q4: How to aggregate attention scores across layers in Figure 1?**
>
> - **A4**: Thank you for the question. Let us clarify: in Δ-AttnMask, we do not aggregate attention scores across layers. As detailed in Section 3.3, our approach is layer-independent. Specifically:
>
>    - We mask hidden states independently for each transformer layer of interest. In the optimized version of Δ-AttnMask, masking is applied only in the second-to-last transformer block.
>
>    - The masking decision for each layer is based solely on the attention scores from that same layer.
>
>    - No cross-layer aggregation is performed, which keeps the method simple and computationally efficient.
>
> - **Q5: How to mask tokens for each layer of VLMs?**
>
> - **A5:** Thank you for the follow-up question. To clarify further:
>
>   Δ-AttnMask does not mask tokens in every layer of the VLM. Instead, we apply masking only in one critical layer—the second-to-last transformer block—where visual and textual features are already deeply fused. This design achieves an optimal balance between performance and computational efficiency, as explained in Section 3.3.
>
>   Implementation details:
>
>   We perform in-place tensor masking via a PyTorch forward hook, which efficiently replaces the hidden states of high-attention tokens with a mask value (e.g., zero or learned token) during the forward pass. This ensures:
>
>    - Minimal overhead—only one additional masking step per sample.
>
>    - Precise targeting of influential tokens without affecting other layers.
>
>    - Easy integration into existing VLM training pipelines.
>
> We will release the full implementation code upon publication to facilitate reproducibility.

---

### Author Response · Authors · 2025-12-04
**General Response**

## General Description:

Dear Area Chairs,

We sincerely thank all Reviewers and Area Chairs for the time and effort dedicated to reviewing our paper. We appreciate the constructive feedback from all reviewers - MExr (R1), SzTa (R2), anGF (R3), ZHFx(R4), which has helped us further improve the presentation and quality of the manuscript. We also thank the reviewers for acknowledging the strengths of our work, including (1) a novel and theoretically grounded idea [R2, R3, R4], (2) a lightweight design and easy to implement [R1, R2, R3, R4], and (3) consistent performance gains across various settings [R1, R2, R3]. The raised concerns are mainly concentrated on (1) more experiments and analyses (R1, R2, R3, R4), and (2) further clarification and discussion (R1, R3, R4).

**Additional Experiments and Analyses:**

In our response, we provide additional experimental results and analyses that address the comments, including:

1. Cross-architecture generalization (R1:Q1 R3:Q8) (Sec. 4.5 in the revised manuscript).
2. Analyses on the masked layers (R3:Q1) (Appendix A.6 in the revised manuscript).
3. Comparative computational analysis (R4:Q3) (Appendix A.4 in the revised manuscript).
4. A discussion of limitations and future directions. (R3:Q3) (Sec. 6 in the revised manuscript).

Moreover, we have clarified specific methodological details and expanded several discussions. These revisions address all reviewer comments in detail.

Thank you again for your insightful feedback and for helping us further refine our work.

Sincerely,

Authors of Submission 7406

---

### Meta-Review · Area_Chair_TGDz · 2026-01-11

**Summary:**

This paper proposes a data selection strategy, AttnMask, for Vision-Language Models (VLMs). Reviewers find this work to be theoretically grounded, lightweight in design, and easy to implement. However, several concerns were raised regarding missing analyses of the masked layers, comparative computational analysis, cross-architecture generalization, and evaluation on larger-scale benchmarks, with the last two concerns still outstanding. Overall, this paper has room to make the evaluation more comprehensive.

**Reviewer Concerns:**

Reviewer MExr

W1: How to derive the unmodified model? [addressed by the rebuttal]

W2: Cross-architecture generalization? [addressed by the rebuttal]

W3: Advantage of the proposed method in mining new, informative data? [addressed by the rebuttal]

W4: How to aggregate attention scores across layers in Figure 1? [addressed by the rebuttal]

W5: How to mask tokens for each layer of VLMs? [addressed by the rebuttal]

---
Reviewer SzTa

W1: Including comparisons with a broader range of baselines. [still outstanding]

R1: The authors claimed that they have compared with state-of-the-art methods, but did not compare with additional baselines as requested.

W2: LLaVA-1.5 665K or comparable datasets? [still outstanding]

R2: It still lacks direct evaluation on LLaVA-1.5 665K or an equivalently large dataset.

W3: The data augmentation results are limited to the small MiniGPT-4 dataset with the Qwen2-VL 2B model. [still outstanding]

R3: No experiments on additional datasets and models are conducted.

---
Reviewer anGF

W1: More related works in this domain need to be discussed. [addressed by the rebuttal]

---
Reviewer ZHFx

W1: The computational cost may be underestimated. [still outstanding]

R1: The scalability concern for very large datasets is not fully analyzed quantitatively.

W2: Including comparisons with a broader range of baselines. [still outstanding]

R2: The authors claimed that they have compared with state-of-the-art methods, but did not compare with additional baselines as requested.

**Reviewer Scores:**

Reviewer MExr: 4 -> 6 (all concerns were addressed, raises the score)

Reviewer SzTa: 4 -> 4 (concerns are still outstanding)

Reviewer anGF: 6 -> 6 (remains a positive rating)

Reviewer ZHFx: 4 -> 4 (concerns are still outstanding)

Average score: 5

---

### Decision · Program_Chairs · 2026-01-26

Reject